# Evaluation of the Enzyme Inhibitory and Antioxidant Activities of *Entada spiralis* Stem Bark and Isolation of the Active Constituents

**DOI:** 10.3390/molecules24061006

**Published:** 2019-03-13

**Authors:** Fatimah Opeyemi Roheem, Siti Zaiton Mat Soad, Qamar Uddin Ahmed, Syed Adnan Ali Shah, Jalifah Latip, Zainul Amiruddin Zakaria

**Affiliations:** 1Department of Pharmaceutical Chemistry, International Islamic University Malaysia, Bandar Indera Mahkota, 25200 Kuantan, Pahang D.M., Malaysia; bukolami_fatty@yahoo.com (F.O.R.); quahmed@iium.edu.my (Q.U.A.); 2Atta-ur-Rahman Institute for Natural Products Discovery (AuRIns), Universiti Teknologi MARA, Bandar Puncak Alam, 42300 Selangor Darul Ehsan, Malaysia; benzene301@yahoo.com; 3Faculty of Pharmacy, Universiti Teknologi MARA, Puncak Alam Campus, 42300 Bandar Puncak Alam, Selangor D.E., Malaysia; 4School of Chemical Sciences and Food Technology, Faculty of Science and Technology, Universiti Kebangsaan Malaysia, 43600 Bandar Baru Bangi, Selangor, Malaysia; jalifah@ukm.edu.my; 5Department of Biomedical Science, Faculty of Medicine and Health Sciences, Universiti Putra Malaysia, 43400 Serdang, Selangor, Malaysia; 6Halal Institute Research Institute, Universiti Putra Malaysia, 43400 Serdang, Selangor, Malaysia

**Keywords:** *Entada spiralis*, 2,2-diphenyl-1-picrylhydrazyl assay, β-carotene assay, 2,2′-azinobis(-3-ethylbenzothiazine-6-sulfonic acid) assay, α-amylase, α-glucosidase, active principles

## Abstract

Digestive enzymes and free radical inhibitors are used to prevent complications resulting from diabetes. *Entada*
*spiralis* (family Leguminosae), which is a well-known medicinal plant in herbal medicine due to its various traditional and medicinal applications, was studied. Crude extracts were successively obtained from the stem bark using petroleum ether, chloroform and methanol as extracting solvents. The antioxidant activity of all the extracts, fractions and isolated compounds were estimated using 2,2-diphenyl-1-picrylhydrazyl (DPPH), β-carotene and 2,2′-azinobis(-3-ethylbenzothiazine-6-sulfonic acid) (ABTS) assays, while digestive enzymes inhibitory activity was assessed using α-amylase and α-glucosidase inhibitory methods. Structure elucidation of pure compounds was achieved through different spectroscopic analysis methods. Fractionation and purification of the most active methanol extract resulted in the isolation of a ferulic ester namely; (e)-hexyl 3-(4-hydroxy-3-methoxyphenyl) acrylate (FEQ-2) together with five known phenolic constituents, identified as kaempferol (FEQ-3), 5,4′-dihydroxy-3,7,3′-trimethoxyflavone (FEQ-2), gallic acid (FEQ-5), (+)-catechin (FEQ-7) and (−)-epicatechin (FEQ-8). FEQ-5 exhibited the strongest antioxidant and enzyme inhibitory activities followed by FEQ-3 and FEQ-4. FEQ-2 also displayed potent free radical scavenging activity with IC_50_ values of 13.79 ± 2.13 (DPPH) and 4.69 ± 1.25 (ABTS) µg/mL, respectively. All other compounds were found active either against free radicals or digestive enzymes.

## 1. Introduction

One of the acceptable ways of treating diabetes is by decreasing post-prandial hyperglycemia [1,2]. The only way to achieve this is by retarding the actions of carbohydrate hydrolyzing enzymes so as to delay the digestion and absorption of glucose through the brush border [3,4]. Commercial enzyme inhibitors such as voglibose, acarbose, miglitol, etc have serious gastrointestinal side effects like diarrhea, flatulent, bloating, etc. [5,6] and this has further increased the search for inhibitors from natural sources. Several plant extracts (mostly polyphenols) have been reported to be powerful starch-hydrolyzing enzyme inhibitors [7,8,9]. These plant-derived inhibitors are more acceptable due to their low cost, and less side effects [10,11]. Furthermore, bioactive components from different plant sources and different groups such as flavonoids, glycosides, phenolic acids, anthocyanins have been reported by different researchers to exert good hydrolyzing enzyme inhibitory activities [12,13,14].

It is assumed that almost 75 percent of the world’s existing plant species have medicinal value and nearly all of these plants possess potent antioxidant potential [15]. This potential is as a result of the presence of both low and high molecular weight secondary metabolites which are derived or synthesized from primary metabolites such as sugars and amino acids through glycosylation, hydroxylation, and methylation [16].

Among all the secondary metabolites phenolic compounds such as flavonoids, flavonoid glycosides, phenolic acids, anthocyanins, tannins, stilbenes and lignans appear to be the most significant as they have been reported to exhibit potent antioxidant activities in various in vivo and in vitro investigations. Compounds with good antioxidant capacity have been found to often demonstrate wide range of biological importance in the treatment of both pathogenic and non-pathogenic diseases including hyperglycemia through inhibition of digestive enzymes [17,18,19]. A notable example among these compounds is montbretin A, a flavonol glycoside which demonstrated potent specific inhibition against human pancreatic amylase (Ki = 8 nM) through intramolecular hydrophobic π-stacking interactions between myricetin and caffeic acid [20].

The genus *Entada adans* (synonym: *Entadopsis* Britton) belongs to a pea family of the Leguminosae and contains approximately twenty-nine species in which *Entada spiralis* is found in Asia. The most common pharmacological property of nearly all species of *Entada* is their antioxidant capacity. Different parts of different species have been reported to exhibit potent antioxidant activity [21,22,23] Similarly, some species have also been reported to exhibit potent digestive enzymes inhibitory activities [24,25].

*Entada spiralis* Ridl. (synonym: *Entada scheffleri*), locally known as Beluru or Sintok, is a woody climber that twins around other higher plants for support and can grow up to 25 m tall. It grows widely in the rain forest and is widely spread across Southeast Asia regions such as Indonesia, Malaysia and Thailand. A previous study on this plant has shown the potency of its methanol extract fraction as an antibactarial and antifungal agent [26]. Compounds isolated so far from this plant include ester saponins, triterpenoidal glycosides, and diterpenoidal glycosides and sugars [27]. Many other pharmacological activities are yet to be properly studied with respect to delving their true medicinal potential particularly on their enzyme inhibitory potential as an antioxidant agent. Hence, this research was aimed to investigate the inhibitory potential of the stem bark on digestive enzymes and free radicals and isolate the active principles responsible for these effects. 

## 2. Results and Discussion

### 2.1. Extraction and Fractionation

Three extracts were obtained from 4.5 kg of powdered material. These are the petroleum ether (22.8 g), chloroform (10.4 g) and methanol (104.6 g) extracts, respectively. The petroleum ether extract is an oily coloured substance with high solubility in non-polar solvents. The chloroform extract on the other hand, appeared as greenish sticky extract, while the methanol extract is dark-brown in colour. Both these extracts are soluble in polar solvents. The differences observed in these extracts could be due to variation and the characteristics of the phytochemical contents present in them. Methanol gave the highest yield, suggesting that most metabolites present in *E. spiralis* stem bark may be polar or moderately polar in nature. Fractionation of methanol extract using CHCl_3_: MeOH gradient elution afforded four pooled fractions, viz. F1 (1.7 g), F2 (1.9 g), F3 (4.8 g) and F4 (12.9 g).

### 2.2. Total Phenolic Contents (TPC)

Table 1 shows the total phenolic and flavonoid contents present in *E. spiralis* extracts and fractions. The methanol extract gave the highest TPC value (42.5 ± 8.59 µg/GAE), as obtained from a gallic acid calibration curve (r^2^ = 0.9941). The TPC values of all *E. spiralis* extracts were observed in increasing order from petroleum ether extract < chloroform extract < methanol extract with the values of 2.6 ± 0.95, 28.3 ± 1.38 and 42.5 ± 8.58 µg/GAE, respectively. Moreover, among all the fractions F1 and F2 were observed to have appreciable and significantly (*p* < 0.05) comparable amount of phenolics (10.71 ± 1.43 and 8.16 ± 3.18 µg/GAE), respectively. F3 had a relatively low TPC with 2.04 ± 3.91 µg/GAE while F4 was the lowest with less than 1 µg/GAE.

### 2.3. Total Flavonoid Contents (TFC)

The methanol extract had the highest TFC with 28.94 ± 2.93 µg QU equivalents/mg dry weight, respectively. The chloroform extract also contained an appreciable amount, with 12.73 ± 1.93 µg, while the petroleum ether extract displayed the lowest content (0.84 ± 0.24 µg QU equivalent/mg dw). F3 contained less than 1 µg of TFC while the result for F4 was negative, suggesting that there was no observable flavonoid content in the fraction. F1 and F2 had the highest and similar TFCs among all the fractions, viz., 6.88 ± 1.20 and 5.12 ± 1.93 µg/mL, respectively. Overall, the TFC values of all the samples were found to be lower than the corresponding TPC ones.

### 2.4. In Vitro Antioxidant Activity 

#### 2.4.1. Dot-Blot Staining Assay Result

Three different detection agents, viz., DPPH^•^, ABTS^•+^ and β-carotene, were used to ascertain the qualitative scavenging activity of the isolated compounds. Radical scavenging activity was indicated by the appearance of yellow spot on a purple background for DPPH^•^, orange/white spots on a green background for ABTS^•+^ and orange spots on a white background for β-carotene. Dot-blot staining results using 1 mg/mL for the isolated compounds from *E. spiralis* stem bark with ascorbic acid as positive control are shown in Figure 1. FEQ-2, FEQ-3, FEQ-5, FEQ-7 and FEQ-8 showed strong intensity on all the three antioxidant detection agents as compared to ascorbic acid which was used as positive control in this assay. FEQ-4 showed strong intensity on ABTS assay, but low intensity on the DPPH and β-carotene assays. 

#### 2.4.2. 2,2-Diphenyl-1-picrylhydrazyl (DPPH) Assay

Analysis of the antioxidant activities of various extracts and fractions at different concentrations (7.81–125 µg/mL) on DPPH radical with their corresponding IC_50_ values is shown in Table 2 with ascorbic acid (AC), Trolox (Tx, for ABTS only) and quercetin (QC) as reference standards. The methanol and chloroform extracts displayed a promising free radical scavenging activity against DPPH^•^. Both extracts displayed a concentration-dependent activity. Methanol extract showed the highest radical inhibitory activity with the lowest IC_50_ value of 42.67 ± 4.10 µg/mL while the petroleum ether one exhibited the lowest activity, with IC_50_ > 1000 µg/mL. The strong free radical scavenging effect of the methanol extract was attributable to the presence of phenolic compounds in significant amounts. Similar activity of their methanol extracts has been previously reported for *E*. *pursaetha* and *E. africana* [28,29]. On the other hand, fractions F1 and F2 obtained from the methanol extract displayed good radical scavenging capacity, with IC_50_ values of 37.09 ± 0.88 and 30.09 ± 1.91 µg/mL, respectively. F3, however, showed inhibition of 78.52 ± 5.21 µg/mL while F4 exhibited the lowest activity, viz., 326.63 ± 13.93 µg/mL. The outcome of this assay showed that F1 and F2 are the most active fractions and therefore isolation of compounds was carried out on these fractions.

Meanwhile, among all the isolated compounds, FEQ-5 demonstrated the most potent activity against DDPH radical with the lowest IC_50_, i.e., 6.15 ± 0.38 µg/mL which was found statistically comparable to the reference standard i.e., ascorbic acid (IC_50_ = 5.38 ± 1.26 µg/mL). Several previous studies have demonstrated structurally similar isolated compounds as good radical inhibitors [30,31]. Similarly, FEQ-2, FEQ-3, FEQ-7 and FEQ-8 exhibited potent DPPH radical scavenging activity (Table 3) with IC_50_ values that are significantly different from one another (13.79 ± 2.13, 11.29 ± 0.26, 12.48 ± 1.55, and 11.75 ± 33.82 µg/mL, respectively). The strong radical scavenging activity of these compounds is in fact a manifestation of the fact that phenolic compounds are well known antioxidants due to their electron-donating capacity [30]. FEQ-4 showed moderate activity (IC_50_ i.e., 66.17 ± 8.03 µg/mL), the outcome of this result corroborated with various spots intensities observed in the dot blot staining assay as described earlier.

#### 2.4.3. 2,2′-Azino-bis (3-ethylbenzothiazoline-6-sulfonic acid) (ABTS) Assay

Unlike the DPPH assay, all the extracts exhibited higher scavenging activity against ABTS^•+^ generated by the reaction between potassium persulphate and ABTS. The activity was observed in a concentration dependent manner. The methanol extract showed the highest activity by displaying the lowest IC_50_ value i.e., 37.39 ± 0.05 µg/mL, followed by the chloroform extract (90.84 ± 3.12 µg/mL) while the petroleum ether extract exhibited the lowest activity with the highest IC_50_ value observed, i.e., 625.16 ± 10.58 µg/mL. F1 efficiently inhibited ABTS radical (19.92 ± 1.11 µg/mL), comparable to that of ascorbic acid (Ac) i.e., 16.74 ± 1.76 µg/mL. F2 and F3 displayed similar activity, with IC_50_ values of 30.50 ± 0.21 and 29.45 ± 4.12 µg/mL, respectively. 

Contrary to the DPPH assay, among all the isolated compounds FEQ-7, FEQ-8 and FEQ-5 showed the high activity, with IC_50_ values i.e., 1.86 ± 0.15, 1.12 ± 0.09, and 1.28 ± 0.89 µg/mL, statistically comparable with one another (Table 3). FEQ-2 and FEQ-3 also demonstrated good ABTS^•+^ scavenging activity having IC_50_ of 4.69 ± 1.25 and 2.10 ± 0.51 µg/mL, respectively. 

### 2.5. In Vitro Digestive Enzymes Inhibitory Activity 

#### 2.5.1. In Vitro α-Amylase Inhibitory Assay

The α-amylase inhibitory activity of *E. spiralis* extracts and methanol fractions using acarbose as positive control were investigated using the starch-iodine method which gives a characteristic blue colour. The intensity of the iodine colour decreases as the starch is hydrolyzed into monosaccharaides by α-amylase. However, in the presence of an active sample, the hydrolysis is forestalled as the colour remains unchanged. The intensity of the blue colour is dependent on the inhibitory capacity of the sample. Enzymes inhibitory activity of the extracts and fractions is shown in Table 2. Chloroform extract and fractions obtained from the methanol extract, viz. F1 and F3, showed a very minimal inhibitory activity with IC_50_ values > 200 µg/mL. The methanol extract showed good inhibition, with an IC_50_ value of 53.12 ± 3.42 µg/mL while F2 was considered the most active, with an IC_50_ of 13.19 ± 0.18 µg/mL. F4 was observed to have negative IC_50_ (-67.45 ± 8.74 µg/mL), revealing the fact that α-amylase enzyme may have been activated rather than being inhibited and therefore could increase the rate of glucose absorption, thereby aggravating post-prandial hyperglycemia in diabetes condition if ingested. However, Only FEQ-3 and FEQ-5 showed moderate inhibitory activity with IC_50_ values of 95.27 ± 3.48 and 87.14 ± 5.31 µg/mL, respectively. FEQ-4 showed weak activity having an IC_50_ close to 500 µg/mL. FEQ-2, FEQ-7 and FEQ-8 did not show any observable α-amylase inhibition. This outcome corroborates the result obtained for the inhibitory activity of the water extract of Qingzhuan dark tea which showed that extracts rich in catechin and epicatechin do not have any significant α-amylase inhibition [32]. The relatively weak enzyme inhibition of FEQ-7 and FEQ-8 could be as a result of non-galloylation and lack of a C2-C3 double bond in conjugation with C=O.

#### 2.5.2. In Vitro α-Glucosidase Inhibitory Assay

The α-glucosidase inhibitory activity of *E. spiralis* extracts and methanol fractions is shown in Table 2. All the extracts and fractions showed different degrees of inhibition in a concentration- dependent manner. The methanol extract, chloroform extract, F1 and F2 showed good α-glucosidase inhibitory activity, having IC_50_ values of less than 100 µg/mL. Meanwhile, the methanol extract, which had the lowest IC_50_ (20.63 ± 0.44 µg/mL) among these samples, was considered the most potent α-glucosidase inhibitor. However, the petroleum ether extract, F3, and F4 exhibited mild α-glucosidase inhibitory activity which can be observed through their high IC_50_ values > 100 µg/mL. The only available similar report on enzyme inhibitory activity of *Entada* species was on the ethanol extract of *E. rheedii* seed coat and pericarp in which a percentage enzyme inhibition of 98.73 ± 0.46 and 74.01 ± 2.02, respectively (1 mg/mL), has been reported [25].

Among the isolated compounds, FEQ-5 showed the highest inhibitory activity (IC_50_, 6.49 ± 1.52 µg/mL) followed by FEQ-3 with IC_50_ of 21.91 ± 3.24 µg/mL as compared to commercial quercetin, reference standard (with IC_50_ 4.85 ± 1.05 µg/mL) as shown in Table 3. FEQ-2 exhibited the lowest inhibition with IC_50_ of 149.00 ± 9.71 µg/mL. FEQ-4 was observed to show moderate inhibitory activity with an IC_50_ value of 75.30 ± 9.83 µg/mL while FEQ-7 and FEQ-8 showed weak inhibition with IC_50_ values of 101.84 ± 7.24 and 99.05 ± 5.20 µg/mL, respectively. Similar weak α-glucosidase inhibitory activity for these compounds with IC_50_ values greater than 200 µM has been reported [33]. Contrary to our findings, FEQ-8 isolated from *Rhodiola crenulata* root was reported to exhibit good inhibitory activity with IC_50_ of 29.85 ± 2.20 µM [34].

### 2.6. Identification of the Isolated Compounds

Six phenolic compounds were successfully isolated from most active fractions of the methanol extract of *E. spiralis* stem bark and their structures elucidated. All the compounds have been isolated for the first time from this plant. FEQ-2 was obtained as a white amorphous powder. Mp: 80–88 °C; UV (λ_max_ MeOH) 365 nm; FT-IR (cm^−1^): 3551, 2914, 2848 1624, 1681 and 1681 ESI-MS [M + H]^+^ ion peak was observed at *m/z* 279.16 (Appendix A) corresponding to C_16_H_22_O_4_. Complete ^13^C [150 MHz] and ^1^H-NMR [600 MHz, CDCl_3_-δ_H_ (ppm)] assignments are given in Table 4. 

All the carbon values were assigned based on the HSQC spectrum. In COSY, protons at δ 1.42 ppm strongly correlated with the terminal primary methyl proton of the fatty acid ester chain at δ 0.90 ppm and methylene protons at δ 1.72 ppm whereas, δ 1.72 ppm was also observed to show correlation with oxymethine protons at δ 4.20 ppm as shown in Figure 2. Similarly, *J*_H_-*J*_H_ correlation was also observed between the two olefinic protons at δ 6.33 and 7.65 ppm. In the aromatic ring, δ 6.95 (H-5) showed a correlation with δ 7.10 (H-6).

In the HMBC spectrum, correlations were observed between the protons at δ 6.33 and δ 7.60 ppm (H-7 and H-8) with δ 127.03 (C-1), 123.03 (C-6), and 167.41 (C=O). These correlations confirmed the attachment of a long fatty acid ester chain and that of the aromatic ring. Moreover, the link between the long methylene chain with COO was established by the correlation between the oxymethylene proton at δ 4.20 ppm with both methylene carbons at δ 31.93 (C-2′), 28.87 (C-3′) and that of C=O at δ 167.42 ppm. Methoxy protons at δ 3.95 ppm displayed strong correlations with the carbon at δ 146.72 suggesting that the methoxy group was attached to C-3. A low intensity broad singlet at δ 5.95 ppm indicated a O-H proton signal which showed a correlation with δ 147.91 (C-4) and δ 114.72 (C-5). Furthermore, the proton at δ 6.95 ppm (H-5) showed a correlation with δ 123.03 (C-6), 127.05 (C-1) and 146.70 (C-4) as illustrated in Figure 2. Based on the interpretation of these spectra data, FEQ-2 was unambiguously identified as hexyl 3-(4-hydroxy-3-methoxyphenyl) prop-2-enoate or hexyl 3-(4-hydroxy-3-methoxyphenyl) acrylate (Figure 3). Though this ferulic ester has previously been synthesized [35], this is the first report of its isolation from the plant kingdom.

FEQ-5 was obtained as white needle-like crystals; mp: 242–244 °C; UV (λ_max_ MeOH) 273 nm; FT-IR (cm^–1^): 3325, 2832, 1656 1448 and 613; ^1^H-NMR [600 MHz, CD_3_OD-δ_H_ (ppm)]: 7.09 (s, 2H, H-2 & H-6) which represents protons at 2 and 6 positions of the aromatic ring. ^13^C-NMR [150 MHz, CD_3_OD, δ_C_ (ppm)]: 121.0 (C-1), 109.0 (C-2 & 6), 145.9 C-3), 138.0 C-4), 145.9 (C-5) 168.0 (C=O). By direct comparison with a previous report [36], FEQ-5, corresponding to C_7_H_6_O_5_, was therefore identified as trihydroxybenzoic acid, also known as gallic acid. It is being reported herein for the first time from the entire *Entada* genus.

Additionally, isolated aglycone flavonoids were also characterized using FT-IR, melting point, 1D-NMR (Table 5 and Table 6) and direct comparison with values previously reported in the literature. The compounds were therefore identified as kaempferol (FEQ-3) [37], pachypodol (FEQ-4) [38], gallic acid (FEQ-5) [36], (+)-catechin [FEQ-7) [39] and (−)-epicatechin (FEQ-8) [39]. The structures of all the isolated compounds are given in Figure 3.

## 3. Materials and Methods

### 3.1. General Information

A Chromatotron model 7924T (T-squared Technology, Inc., Sneath Lane, San Bruno, CA, USA) and chromatography columns of different sizes (30, 20 and 10 mm) purchased from Kemtech (Pleasant Prairie, WI, USA) were used for purification of the constituents. A FTIR spectrometer (Perkin Elmer Inc., Hopkinton, MA, USA) equipped with horizontal attenuated total reflectance device with diamond crystal and Perkin Elmer spectrum software version (10.03.09) was used to detect the functional groups of the isolated constituents. The mass spectrum of the new isolated compound was determined using an Agilent 1290 Infinity LC system coupled to an Agilent 6520 accurate-Mass Q-TOF mass spectrometer (Agilent Technologies, Santa Clara, CA, USA). An 1800 series UV-VIS spectrophotometer (Shimadzu, Kyoto, Japan) was used to detect the presence or absence of chromophores. Melting points were measured using an SMP 10 apparatus, (ST50SA Bibby Sterilin, Ltd., Stone, UK). ^1^H-, and ^13^C- NMR spectra (600 and 150 MHz, respectively) were measured using a FT-NMR cryoprobe Bruker Advance 111 spectrometer (Bruker Scientific Technology Co., Ltd., Yokohama, Japan). Absorbance was measured using a TECAN PRO 200 microplate reader (Tecan Trading AG, Mannedorf, Switzerland). A rotary evaporator (IKA RV 10B S99, 40 °C, 115 rpm) (Buchi, Flawil, Switzerland) was used to concentrate the extracts. Aluminium chloride (AlCl_3_), 2,2 diphenyl-1-picrylhydrazyl (DPPH), 2,2′-azinobis-3-ethylbenzothiazine-6-sulfonic acid (ABTS), sodium carbonate (Na_2_CO_3_), *p*-nitrophenyl glucopyranoside (*p*NPG), potassium phosphate and 6-hydroxy-2,5,7,8-tetramethylchroman-2-carboxylic acid (Trolox) were purchased from Sigma-Aldrich (St. Louis, MO, USA). Gallic acid, Folin-Ciocalteu reagent (FC) and Whatman No 1 filter paper were purchased from GE Healthcare UK Limited (Amersham, UK). Silica gel 60 (0.2–0.5 mm and 0.063–0.200 mm), Pf254 with gypsum, preparative TLC (20 × 20 cm) plates were obtained from Merck (Darmstadt, Germany). All other chemicals and solvents used were of analytical reagent quality.

### 3.2. E. spiralis Stem Bark Preparation and Extraction

*E. spiralis* Ridl. stem barks were obtained from Tasik Chini Forest, Pekan District, Pahang, Malaysia. (voucher specimen KMS-5228) and cut into smaller pieces, air-dried at room temperature, and pulverized into powdered form to give a final mass of 4.5 kg. The powder was then macerated successively using solvents of different polarity (petroleum ether, chloroform, and methanol) to obtain petroleum ether, chloroform, and methanol extracts, respectively. Maceration with each solvent was repeated until exhaustion before proceeding to the next solvent and the resultant filtrates from each solvent were concentrated in vacuo using a rotary evaporator. The crude extracts were packed in a glass bottle and kept in the fridge until further analysis.

### 3.3. Fractionation of the Active Extract

Being the most active the methanol extract (100 g) was fractionated on a silica gel column with gradient mixtures of chloroform: MeOH (9:1→6:4) as eluent to afford four pooled fractions, viz. F1 (1.7 g), F2 (1.9 g), F3 (4.8 g) and F4 12.9 g. Each fraction was concentrated in vacuo and kept in a glass bottles and in the fridge until further analysis. All four fractions were evaluated for bioactivity and the most active fractions were subjected to different purification processes to obtain their active principles. 

### 3.4. Estimation OF Total Phenolic Content (TPC)

The total phenolic contents in *E. spiralis* extracts and fractions were determined using a Folin-Ciocalteu (FC) method adapted and modified from Sulaiman et al. [40]. Briefly, 50 µL of 10% FC *w*/*v* was introduced into flat-bottom microplate wells followed by 10 µL of standard or sample (7.18–1000 µg/mL) against blank containing only the methanol. Finally, 50 µL of 40% Na_2_CO_3_ was then added to each and incubated for 2 h at room temperature. Gallic acid was used as standard. Assay was conducted in triplicate. Absorbance was measured at 725 nm using microplate reader and the total phenolic content was determined from the linear regression curve of absorbance against concentration using the equation Y = mx + c. Results obtained were expressed as microgram gallic acid equivalent (µg of GAE/mg/dw of extract).

### 3.5. Estimation of Total Flavonoid Contents (TFC)

Total flavonoid content in *E. spiralis* extracts and fractions were measured using methods adapted from Abdel-Hameed at al. and Ahmed et al. [41,42] with modifications. Briefly, 100 µL of 2% AlCl_3_ dissolved in methanol was added to same amount of extract (1 mg/mL) or standard (7.81–250 µg/mL). Quercetin was used as standard. Blanks contained extracts with methanol only without AlCl_3_. Absorbance was noted at 415 nm after 15 min. The tests were conducted in triplicate and quercetin calibration curve was used to determine the concentration of each extract using the equation Y = mx + c. Results were expressed as milligram of quercetin equivalence per mg/dw of the sample.

### 3.6. In Vitro Antioxidant Activity Assay

#### 3.6.1. Dot-Blot Staining Assay 

Rapid detection of the free radical scavenging activity of isolated compounds from active fractions of *E. spiralis* methanol extract was carried out using a method adapted from Sarian et al. [43]. Three detection agents viz., DPPH^•^, ABTS^•+^ and β-carotene were used. Briefly, few drops of each compound were spotted on a TLC plate (1.5 × 8 cm) using capillary tube and dried for few minutes at room temperature. The plate was then carefully dipped in 0.5% DPPH^•^ in methanol for few seconds. The plate was allowed to dry and the colour change was observed. Yellow spots on a purple background indicate radical scavenging activity and intensity of the yellow colour is proportional to the antioxidant activity. A TLC-β-carotene assay was also conducted using 0.05% of β-carotene in chloroform. A few drops of each sample (1 mg/mL) was spotted on TLC plate an allowed to dry after which it was dipped into β-carotene solution for 20 s. The plate was dried completely under direct sunlight for 3 h. Yellow or orange spot on a white background indicates activity. The intensity of the colour is dependent on the extent of antioxidant activity of the compound. TLC ABTS^•+^ test was performed by mixing 2.45 mM of potassium persulphate with 7 mM of ABTS and kept in the dark for 16 hours to generate ABTS^•+^. Already spotted TLC plates were then dipped into it for 10 s and allowed to dry at room temperature. White or orange spots on a bluish green background indicates activity.

#### 3.6.2. 2,2-Diphenyl-1-picrylhydrazyl (DPPH) Assay

Modified methods from Sulaiman et al. and Ahmed et al. [40,42] were used. Briefly, 150 µL of freshly prepared DPPH^•^ solution (0.4 M concentration) was carefully introduced into 96 round microplate wells each. 100 µL of sample/ standard (7.81–500 µg/mL) were added. Blanks contained only methanol and DPPH^•^. Ascorbic acid (dissolved in distilled water) and quercetin (dissolved in MeOH) were used as standard. The plate was left in the dark for 25 min to activate. Absorbance was noted at 517 nm. Tests were conducted in triplicate and the percentage inhibition of each sample/standard was calculated using the following equation:(1)$$\left(\%\;\mathrm{DPPH}\;\mathrm{inhibition}\right)=\;[\frac{Ac-As}{Ac}]\;\times 100$$
where, *Ac* = absorbance of DPPH radical in MeOH, *As* = absorbance of DPPH radical in sample or standard. IC_50_ values obtained from graphical plot from percentage inhibition against concentration were used to define the radial scavenging activity of each sample.

#### 3.6.3. 2,2′-Azino-bis(3-ethylbenzothiazoline-6-sulfonic acid) (ABTS) Assay

A method reported by Zheleva-Dimitrova et al. [44] was adopted with slight modifications. Stock solutions of ABTS (7 mM) and potassium persulphate (2.45 mM) were prepared using distilled water. The working solution was prepared by adding 1 mL of ABTS solution to same amount of potassium persulphate solution. The reaction mixture was left overnight for 16 h to generate the intensely blue coloured ABTS^•+^. 1 mL of ABTS^•+^ was then added to 50 mL of MeOH and distributed (100 µL each) into a 96 microplate well containing 100 µL of serially diluted sample/ standard (7.81–125 µg/mL). Absorbance was measured at 734 nM against blank (containing sample and MeOH only). Test was conducted in triplicate. Samples were compared with trolox and ascorbic acid and percentage inhibition was calculated as follows:(2)$$\left(\%\;\mathrm{ABTS}\;\mathrm{inhibition}\right)=\;[\frac{Ac-As}{Ac}]\;\times 100$$
where, *Ac* = absorbance of ABTS^•^ in MeOH, *As* = absorbance ABTS^•+^ in sample/standard. 

The radical scavenging activity was determined from the IC_50_ values obtained from the percentage inhibition curve against different concentrations of the sample/standard.

### 3.7. In Vitro Digestive Enzymes Inhibitory Activity

#### 3.7.1. In Vitro α-Amylase Assay Inhibitory Assay

This assay measured the hydrolysis of starch in the presence of α-amylase, quantified by iodine which shows a blue-black colour with starch. The procedure proposed by Johnson et al. [45] was adopted and modified for a microplate-based method carried out by Xiao et al. [46]. Briefly, 25 µL of α-amylase solution (1 U/mL in 0.2 M phosphate buffer at pH 7) was introduced into 96 well flat bottom microplates followed by addition of 25 µL of serially diluted sample/standard (7.8–1000 µg/mL) and incubated for 5 min at 37 °C followed by 25 µL of 1% starch in buffer (boiled for 15 min) and re-incubated for 45 min at 50 °C. Next was the addition of 100 µL of 1% iodine solution and 50 µL of distilled water. The microplate was tightly covered during incubation to prevent evaporative loss. Absorbance was noted at 565 nm. All determinations including sample, control, and blank were conducted in triplicate under the same reaction conditions. Acarbose was used as standard. Inhibitory activity was determined using the following equation:(3)$$\left(\%\;\mathrm{enzyme}\;\mathrm{inhibition}\right)=\;1-[\frac{Abs}{Contr}]\;\times 100$$
where, *Abs* is the absorbance of the sample or standard. *Contr* is the absorbance of the control without enzyme.

#### 3.7.2. In Vitro α-Glucosidase Inhibitory Assay

This assay used a method adapted from Johnson et al. [45] with slight modifications employing α-glucosidase (enzyme) obtained from *Saccharomyces cerevisiae* (1 U/mL) dissolved in 50 mM phosphate buffer (pH 6.9). 100 µL of enzyme (in 0.1M phosphate buffer) was added into 96 microplate well containing 50 µL of sample/ standard 7.8–1000 µg/mL and incubated at 37 °C for 10 min. Next was the addition of 50 µL of the substrate, *p*-nitrophenyl glucopyranoside (5 mM in 0.1 phosphate buffer at pH 6.9) and re-incubated for 5 min at 37 °C. The reaction was terminated by the addition of 100 µL of 0.1 M Na_2_CO_3_ and absorbance was measured at 405 nm. Quercetin was used as standard. The percentage inhibition of α-glucosidase was calculated using the following equation:(4)$$\left(\%\;\mathrm{enzyme}\;\mathrm{inhibition}\right)=\;1-[\frac{s-b}{c}]\;\times 100$$
where, *s* = absorbance of the sample/standard, *b* = absorbance of the blank containing no substrate, *c* = control containing buffer in place of sample.

The percentage of the extract required to inhibit 50% of the α-glucosidase activity (IC_50_) was determined from the regression curve. Experiments were conducted in triplicate.

### 3.8. Active Principle Isolation through Column Chromatography

Active fraction F2 (1.9 g) was subjected to silica gel column chromatography and eluted with gradient of Hex: CHCl_3_ 50:50→0:100 and CHCl_3_: MeOH 90: 10→50: 50 to afford fractions F2A (150 mg), F2C (62 mg), F2D (51 mg), F2H (78.4 mg), F2G (118.2 mg). Fraction F2A was re-chromatographed on a silica gel open column with CHCl_3_: EA 90:10→50:50 as eluent. Fraction F2A_1–18_ (82.6 mg) was further purified on flash column using CHCl_3_: EA 60:40 isocratic to afford FEQ-4 (10 mg). F2H was dissolved in a small quantity of chloroform followed by the addition of hexane to form a whitish precipitate which was filtered off and washed several times with hexane. The precipitate obtained (33 mg) was then dissolved in chloroform and adsorbed on silica gel (0.623–0.400 mm). Fraction F2C (62 mg) was eluted with 40%, 50% and 60% PET: DCM on a silica gel open column. Purification of the 60% part on a preparative TLC plate with PET: DCM (50:50) afforded FEQ-2 (15.0 mg). Fraction F2D (51 mg) was dissolved in methanol. The methanol-insoluble part was filtered off while the filtrate (33 mg) was purified on a preparative TLC plate using CHCl_3_: MeOH (60:40) as developing solvent to afford FEQ-7 (6.3 mg). Fraction F_2_G (118.2 mg) was subjected to silica gel column chromatography eluting with CHCl_3_: EA 90:10→0: 100 to afford combined fractions F2G_19–24_ (44 mg), F2G_7–18_ (38.5 mg) and F2G_28–36_ (8.5 mg). Fraction F2G_19–24_ was repeatedly purified on preparative TLC plates with EA: FA (90:10) to yield FEQ-8 (5.0 mg). Fraction F2G_7–18_ was eluted with EA: MeOH 100:0→50:50 on a silica gel open column. Fractions collected at 80:20 and 70:30 (22 mg) were combined and re-purified on a flash column using the same solvent system to yield FEQ-5 (9.2 mg). 

### 3.9. Isolation Using Chromatotron

Active fraction F1 (1.7 g) was dissolved in methanol to remove the methanol-insoluble (102.3 mg) part by filtration. The filtrate (900 mg) was subjected to centrifugal chromatography with 4 mm rotor with gradient system of DCM: CHCl_3_ 90:10→0: 100 to afford F1_1–12_ (78 mg) and F1_15–23_ (43 mg). Band separation was observed with the aid of UV light. F1_1–12_ (78 mg) was re-chromatographed using 1 mm rotor and eluted with 80% DCM:20% CHCl_3_ to afford FEQ-3 (4.1 mg). 

### 3.10. Statistical Analysis

Data obtained were presented as mean ± SEM of three replicate measurements. Results were analyzed using the SPSS software version 20 (IBM, United Kingdom Limited, North Harbour, UK) with analysis of variance (ANOVA). Correlation between parameters was performed using Pearson’s coefficient. Significance was determined at *p* < 0.05.

## 4. Conclusions

Results obtained through our research work further validate *E. spiralis* stem bark’s methanol extract as a potent free radical and digestive enzymes inhibitor. Repeated fractionation and purification of the active methanol extract of *E. spiralis* stem bark has resulted in the isolation of a novel ferulic ester together with five biologically active phenolic compounds, comprising a phenolic acid and four flavonoids. All these compounds were found active either as free radical quenchers or/and digestive enzyme inhibitors. Interestingly, this is the first scientific report on the isolation of kaempferol, catechin and epicatechin from *E. spiralis* Ridl. To the best of our knowledge, pachypodol and gallic acid have also been isolated from the entire *Entada* genus for the first time in this study.

## Figures and Tables

**Figure 1 molecules-24-01006-f001:**
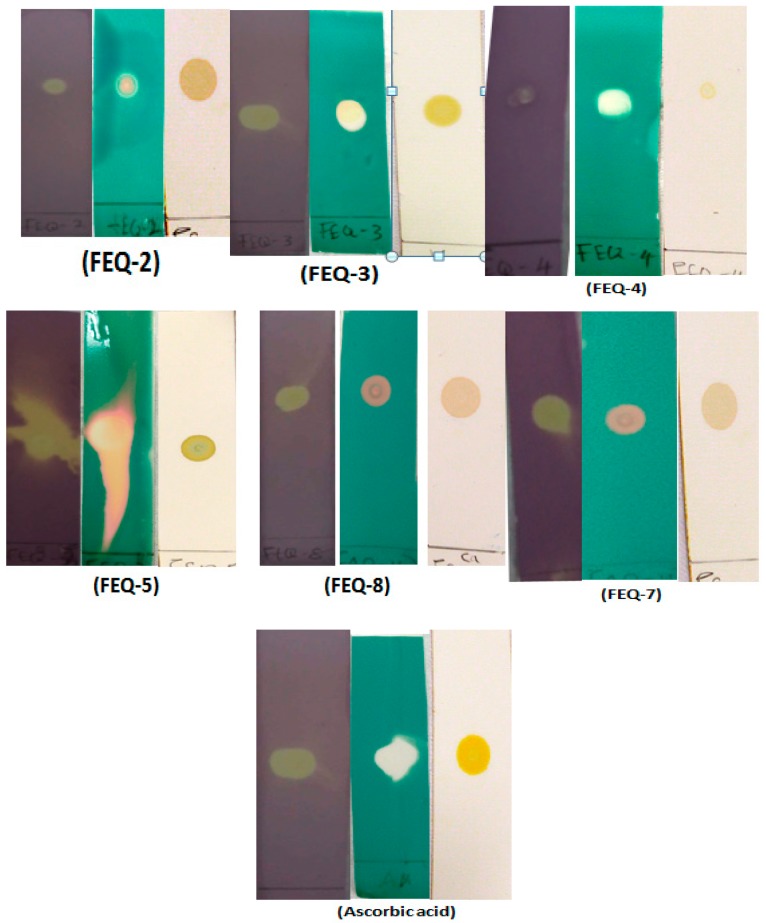
Result of the Dot-blot staining assay for the isolated compounds. Purple-background plate (plate sprayed with DPPH reagent), blue background plate (plate sprayed with ABTS reagent), white background plate (plate sprayed with β-carotene).

**Figure 2 molecules-24-01006-f002:**
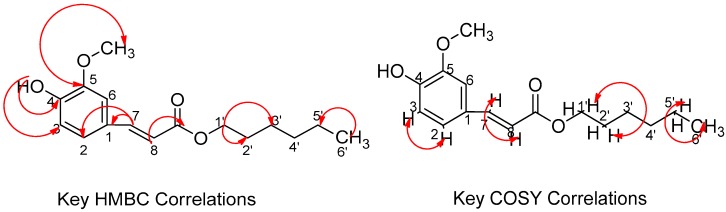
Significant HMBC and ^1^H-^1^H COSY Correlations for FEQ-2 isolated from the methanol extract of *E. spiralis* stem bark.

**Figure 3 molecules-24-01006-f003:**
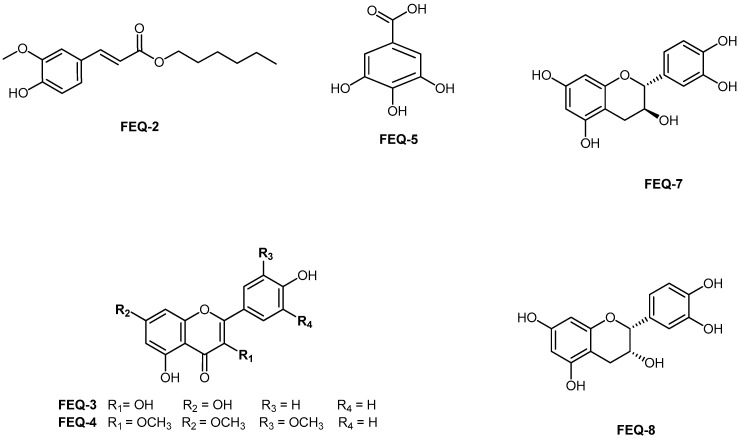
Structure of phenolic constituents from *Entada spiralis* methanol extract.

**Table 1 molecules-24-01006-t001:** Estimation of total phenolic and flavonoid contents of various extracts and fractions of *E. spiralis* stem bark.

Extract	TPC (µg GAE/mg dw)	TFC (µg QUE/mg dw)
Methanol Extract	42.5 ± 8.59 ^e^	28.94 ± 2.93 ^e^
Chloroform Extract	28.3 ± 1.38 ^d^	12.73 ± 1.93 ^d^
Petroleum Ether Extract	2.6 ± 0.95 ^b^	0.84 ± 0.24 ^b^
F1	10.71 ± 1.43 ^c^	6.88 ± 1.20 ^c^
F2	8.16 ± 3.18 ^c^	5.12 ± 1.93 ^c^
F3	2.04 ± 3.91 ^b^	0.51 ± 0.99 ^b^
F4	0.79 ± 1.23 ^a^	−468 ± 21.55 ^a^

Values are expressed as mean ± SEM (*n* = 3) of triplicate measurements. Values with different letter on the same column are significantly different (at *p* < 0.05) as measured by Turkey’s HSD test. TFC (total flavonoid content), GAE (gallic acid equivalence), QUE (quercetin equivalence), TPC (total phenolic content).

**Table 2 molecules-24-01006-t002:** Radical scavenging and enzyme inhibitory activity of extracts and methanol fractions from *Entada spiralis* stem bark.

Sample	IC_50_ (µg/mL)			
	DPPH Assay	ABTS Assay	α-Amylase	α-Glucosidase
Methanol extract	42.67 ± 4.10 ^C^	37.39 ± 0.05 ^C^	98.15 ± 6.04 ^C^	20.63 ± 0.44 ^A^
Chloroform extract	472.83 ± 11.20 ^F^	90.84 ± 3.12 ^D^	405.29 ± 7.36 ^F^	74.96 ± 24.77 ^C^
Petroleum Ether extract	1050.08 ± 23.21 ^G^	625.16 ± 10.58 ^F^	-47.60 ± 9.25 ^G^	172.93 ±1.77 ^F^
F1	37.09 ± 0.88 ^C^	19.92 ± 1.11 ^A^	312.14 ± 8.78 ^E^	24.17 ± 1.24 ^B^
F2	30.18 ± 1.91 ^B^	30.50 ± 0.21 ^B^	13.19 ± 0.18 ^B^	28.15 ± 2.25 ^B^
F3	78.52 ± 5.21 ^D^	29.45 ± 4.12 ^B^	215.86± 16.62 ^D^	123.18 ± 2.05 ^D^
F4	326.63 ± 13.93 ^E^	303.75 ± 12.29 ^E^	−67.45 ± 8.74 ^H^	143.76 ±21.43 ^E^
Qc	29.82 ± 3.73 ^A^	-	-	4.85 ± 1.05
Ac	24.67 ± 0.45 ^A^	16.74± 1.76 ^A^	-	-
Acarbose	-	-	0.85 ± 0.19	-
Tx	-	15.23 ± 2.15 ^A^		

Values are expressed as mean ± SEM of triplicate measurement. Samples were analyzed using one-way ANOVA. Values of different samples with similar superscript are not different significantly as measured by Turkey’s HSD post Hoc test (at *p* < 0.05). Ac (ascorbic acid), Tx (Trolox), Qc (quercetin), ND (not detected), IC_50_ (concentration of the sample required to scavenge 50% of the radicals).

**Table 3 molecules-24-01006-t003:** Radical scavenging and enzyme inhibitory activity of isolated compounds from *Entada spiralis* stem bark.

Sample	IC_50_ (µmol/L)			
	DPPH Assay	ABTS Assay	α-Amylase	α-Glucosidase
FEQ-2	5.0 × 10^−2^	1.0 × 10^−2^	ND	5.3 × 10^−1^
FEQ-3	3.9 × 10^−2^	7.0 × 10^−3^	3.3 × 10^−1^	7.7 × 10^−2^
FEQ-4	1.9 × 10^−1^	1.0 × 10^−1^	1.4 × 10^0^	2.2 × 10^−1^
FEQ-5	3.6 × 10^−2^	7.5 × 10^−3^	5.1 × 10^−1^	3.8 × 10^−2^
FEQ-7	4.3 × 10^−2^	6.4 × 10^−3^	ND	3.5 × 10^−1^
FEQ-8	4.0 × 10^−2^	3.9 ×10^−3^	ND	3.4 × 10^−1^
Qc	9.9 × 10^−2^	−	−	1.6 × 10^−2^
Ac	1.4 × 10^−1^	9.5 × 10^−2^	−	−
Acarbose	−	−	1.3 × 10^−3^	−
Tx	−	6.1 × 10^−2^		

Values are expressed as mean ± SEM of triplicate measurement. Samples were analyzed using one-way ANOVA. Ac (ascorbic acid), Tx (Trolox), Qc (quercetin), ND (not detected), IC_50_ (concentration of the sample required to scavenge 50% of the radicals). FEQ-2 (hexyl ferulate), FEQ-3 (kaempferol), FEQ-4 (pachypodol), FEQ-5 (gallic acid), FEQ-7((+)-catechin), FEQ-8 ((−)- epicatechin).

**Table 4 molecules-24-01006-t004:** ^1^H-NMR (600 MHz) and ^13^C-NMR (150MHz) in CDCl_3_ spectra data of FEQ-2.

Position	δ_H_ (ppm), m J (Hz)	δ_C_ (ppm) Type
1	-	127.06, C
2	7.10 1H, dd (6.8, 1.4)	109.32, CH
3	6.95 d, (6.8)	146.78, CH
4	-	147.92, C
5	-	114.72, C
6	7.06 1H, d (1.5)	123.04, CH
7	7.60 1H, d (15.9)	144.65, CH
8	6.33 1H d (15.9)	115.67, CH
1′	4.20 2H, t (6.8)	64.64, OCH_2_
2′	1.63–1.73, 2H, m	31.93, CH_2_
3′	1.63–1.73, 2H, m	28.87, CH_2_
4′	1.35–1.43, 2H, m	29.71, CH_2_
5′	1.35–1.43, 2H, m	26.06, CH_2_
6′	0.90 3H, t (7.14)	14.11, CH_3_
OCH_3_	3.95 3H, s	55.93, OCH_3_
C=O	-	167.42
O-H	5.96 1H, s	-

**Table 5 molecules-24-01006-t005:** Identification of the aglycone flavonoids isolated from the methanol extract of *E. spiralis* stem bark.

Compound	MP (°C)	UV (λ_max_ MeOH)	FT-IR (cm^−1^)	MF	Compound’s Name
FEQ-3	278–280	214, 281	3291, 2924, 1257, 1613	C_15_H_10_O_6_	Kaempferol
FEQ-4	170–172	225, 256	3260, 2938, 1606, 1250	C_18_H_17_O_7_	Pachypodol
FEQ-7	174–178	236, 280	3325, 2832, 1647, 1448	C_15_H_15_O_6_	(+)-Catechin
FEQ-8	241–244	280, 238	3325, 2883, 1633, 1448	C_15_H_15_O_6_	(−)-Epicatechin

MF (molecular formula), MP (melting point).

**Table 6 molecules-24-01006-t006:** ^1^H, ^13^C-NMR chemical Shifts (ppm) of the four aglycone flavonoids isolated from *E. spiralis* methanol extract.

Position	FEQ-3		FEQ-4		FEQ-7		FEQ-8	
	δ_H_, m J (Hz)	δ_C_	δ_H_, m J (Hz)	δ_C_	δ_H_, m J (Hz)	δ_C_	δ_H_, m, J (Hz)	δ_C_
2	-	146.7	-	156.5	4.49 (d) (*J* = 7.5)	81.5	4.71 (m)	78.5
3	-	135.7	3.81 (s)	138.7	3.84 (dd) (*J* = 13.2, 7.6)	66.8	3.82 (m)	65.4
4	-	176.6	-	178.8	2.38(m) 2.68 (dd) (*J* = 6.0, 5.3)	28.3	2.49 (dd) (*J* = 16.4, 4.4), 2.66 (m)	28.7
5	-	161.1	-	161.3	-	156.6	-	157.0
6	6.21 (d) (*J* = 2.0)	97.9	6.33 (d) (*J* = 2.2)	97.7	5.90 (d) (*J* = 2.0)	95.6	5.90 (d) (*J* = 1.9)	95.6
7	-	164.3	3.89 (s)	165.8	-	156.9	-	156.7
8	6.42 (d) (*J* = 2.04)	93.1	6.55 (d) (*J* = 2.2)	91.9	5.70 (d) (*J* = 2.1)	94.3	5.73 (d) (*J* = 1.9)	94.6
9	-	156.9	--	156.9	-	155.8	-	156.2
10	-	103.1	-	105.9	-	99.5	-	99.0
1′	-	122.4	-	122.7	-	131.1	-	131.1
2′	8.12 (dd) (*J* = 2.0, 7.0)	129.3	7.65 (m)	110.9	6.70 (d) (*J* = 2.0)	115.0	6.69 (m)	115.4
3′	6.94 (dd) (*J*= 2.0, 9.0)	115.0	3.97 (s)	146.1	-	145.3		144.9
4′	-	159.2		150.3	-	145.3	-	145.0
5′	6.94 (dd) (*J*= 2.0, 9.0)	115.0	7.03 (d) (*J* = 8.4)	114.9	6.73 (d) (*J* = 1.8)	115.5	6.90 (s)	115.2
6′	8.12 (dd) (*J* = 2.0, 7.0)	129.3	7.65 (m)	120.9	6.60 (dd) (*J* = 8.1, 1.9)	118.9	6.69 (m)	118.4

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
