# Peer review of "Evaluation of the Enzyme Inhibitory and Antioxidant Activities of Entada spiralis Stem Bark and Isolation of the Active Constituents"

_molecules, 2019, doi:10.3390/molecules24061006_

Reviewer 1 Report

The authors have performed an accurate isolation and identification of the phenolic compounds present in Entada spiralis stem bark, with a proper methodology based on NMR and IR.

Some points remained to be clarified:

-          Fractions 1, 2, 3 and 4 show different TPC and TFC composition and in vitro activities. It is not clear the different fractionation process applied to obtain them. Paragraph 291-296 explains that gradient mixtures of chloroform : methanol were used and gives the g obtained for each fraction. Please clarify which is the mixture of solvents used for each fraction and the meaning of the g in parentheses.

 -          Similarly, it is important to clarify that the differences observed in the fractions and extracts are due to the different hydrophilic and lipophilic characteristics of the phenolic compouds, that are extracted better in one solvent compared to another. These polar-less polar properties should be stated somewhere.

 -          It is surprising that F4 has a negative effect in the in vitro amylase inhibitory assay, it might be due to the nule content on phenolics and hence it behaves as a “blank”, as a pure solvent, rather than provoke a real inhibition on the enzyme activity.

Minor changes of typing errors:

-          Line 137: it should say petroleum ether extract < chloroform extract < methanol extract

-          Line 138: it should say 42.5 ± 8.58 µg/GAE

Author Response

1.      Fractions 1, 2, 3 and 4 show different TPC and TFC composition and in vitro activities. It is not clear the different fractionation process applied to obtain them. Paragraph 291-296 explains that gradient mixtures of chloroform : methanol were used and gives the g obtained for each fraction. Please clarify which is the mixture of solvents used for each fraction and the meaning of the g in parentheses.

 RESPOND:

·         The fractionation process used has been stated in the text. Please refer to Page 4-5, lines 314-319.

·         The fractions F1, F2, F3 and F4 were obtained based on the similarity in the TLC profile of the eluate obtained from the solvent mixtures of chloroform: methanol 9:1 to 6:4.

·         “g” means gram

2.      Similarly, it is important to clarify that the differences observed in the fractions and extracts are due to the different hydrophilic and lipophilic characteristics of the phenolic compouds, that are extracted better in one solvent compared to another. These polar-less polar properties should be stated somewhere.

 RESPOND:

·         Additions have been made. Please see Page 4, lines 45-51. Changes highlighted in purple

3.      It is surprising that F4 has a negative effect in the in vitro amylase inhibitory assay, it might be due to the nule content on phenolics and hence it behaves as a “blank”, as a pure solvent, rather than provoke a real inhibition on the enzyme activity.

 RESPOND:

·         Yes, F4 has a negative effect , this could be attributed to low phenolic content in the fraction.

4.      Line 137: it should say petroleum ether extract < chloroform extract < methanol extract

 RESPOND:

Changes have been effected. Please see Page 4, lines 58-59. Changes highlighted in yellow

5.      Line 138: it should say 42.5 ± 8.58 µg/GAE

 RESPOND:

Error has been corrected. Please refer to Page 4, line 59. Changes highlighted in yellow

Reviewer 2 Report

       In this manuscript, entitled “Evaluation of Enzyme Inhibitory and Antioxidant Activities of Entada spiralis Stem Bark and Isolation of the Active Constituents”, the authors investigated the inhibitory potential of some isolated fractions from the E. spiralis stem bark on digestive enzymes and free radicals.  The authors have reported that gallic acid (FEQ-5) in the methanol extract exhibited the strongest antioxidant and enzyme inhibitory activities.  An isolated ferulic ester (FEQ-2) fraction displayed potent free radical scavenging activity with different IC50 values.

       The results shown in this manuscript were not sufficient to support the judgment and claims in both the discussion and concluding remarks.  Some major comments are listed below:

1)  The active principles of fraction F2 and F1, namely FEQ-2, FEQ-3, FEQ-4, FEQ-5, FEQ-7 and FEQ-8, isolated from methanol extract of E. spiralis, have been tested for bioactivity in this manuscript.  Please double check why the pure compound FEQ-6, isolated from fraction F2 of methanol extract, has been omitted in the manuscript?

2)  In the text, it was stated that IC50 > 200 μg/mL was regarded as a very minimal inhibitory activity.  As indicated in Table 5, the statement in both the abstract and conclusions “All these compounds were found active either as free radical quenchers or/and digestive enzymes inhibitors.” were not supported by the results.

3)  Among the isolated compounds, most of them are common antioxidative phenolic ingredients (e.g. gallic acid, catechin, epicatechin).  As stated in the introduction, some other compounds (e.g. ester saponin, triterpenoidal glycosides, and diterpenoidal glycosides) have been isolated from this plant in other literatures.  Why none of these active compounds (other than the common phenolic compounds) have been found and mentioned in this sample?

4)  Since the manuscript was aimed to study the potential pharmacological activities (i.e. to prevent diabetes complications) of this plant, only preliminary antioxidant activity and inhibitory activity on amylolytic enzymes have been addressed in the present study.  The existence of phenolic compounds and their antioxidative activities of a particular plant might not necessarily be offering pharmacological activities.  Hence, it seems that this work is still preliminary and additional relevant experiments will be suggested to have a step forward work on the potential pharmacological activities in preventing diabetes complications as mentioned in the abstract.

5)  The interventional timing of each sample should be elaborated in the method of in vitro α-amylase inhibitory assay, in vitro α-glucosidase inhibitory assay, and ABTS assay.

6)  Why IC50 (μg/mL) was chosen as an approach to determine the inhibitory activity on digestive enzyme, but not the direct magnitude of reduction of enzyme activity?

7)  The selection criteria of fractionation among methanol extract, chloroform extract, and petroleum ether extract should be explained and mentioned in the manuscript.

8)  The reason of selecting only F1 and F2 fractions to obtain the isolated active ingredients should be explained in the text.

9)  In Table 5: please double check the value “215 ± 86 ± 16.62”

10) Extensive format check is needed throughout the manuscript.

Author Response

1.      The active principles of fraction F2 and F1, namely FEQ-2, FEQ-3, FEQ-4, FEQ-5, FEQ-7 and FEQ-8, isolated from methanol extract of E. spiralis, have been tested for bioactivity in this manuscript.  Please double check why the pure compound FEQ-6, isolated from fraction F2 of methanol extract, has been omitted in the manuscript?

 RESPOND:

·         It was typo. The sentence that explained the isolation of FEQ-6 has been omitted. Please see Page 18.

2.      In the text, it was stated that IC50 > 200 μg/mL was regarded as a very minimal inhibitory activity.  As indicated in Table 5, the statement in both the abstract and conclusions “All these compounds were found active either as free radical quenchers or/and digestive enzymes inhibitors.” were not supported by the results.

 RESPOND:

·         Chnages has been made as follows.“All these compounds were found active either as free radical quenchers or/and digestive enzymes inhibitors.” means that some compounds are active on both free radicals and enzymes tested e.g FEQ-3 and FEQ-5 while some are only active on free radicals and α-glucosidase enzyme.

3.      Among the isolated compounds, most of them are common antioxidative phenolic ingredients (e.g. gallic acid, catechin, epicatechin).  As stated in the introduction, some other compounds (e.g. ester saponin, triterpenoidal glycosides, and diterpenoidal glycosides) have been isolated from this plant in other literatures.  Why none of these active compounds (other than the common phenolic compounds) have been found and mentioned in this sample?

 RESPOND:

·         Yes all the phenolic compounds isolated have been previously reported except FEQ-2. However, this is the first time of reporting the isolation of these compounds from E. spiralis stem bark. Other compounds such as ester saponin, triterpenoidal glycosides, and diterpenoidal glycosides have been previously isolated from this plant as stated in the text but concentrated on the isolation of phenolic compounds especially flavonoids. Please see Page 3, lines 35-36, Reference number 26

4.      Since the manuscript was aimed to study the potential pharmacological activities (i.e. to prevent diabetes complications) of this plant, only preliminary antioxidant activity and inhibitory activity on amylolytic enzymes have been addressed in the present study.  The existence of phenolic compounds and their antioxidative activities of a particular plant might not necessarily be offering pharmacological activities.  Hence, it seems that this work is still preliminary and additional relevant experiments will be suggested to have a step forward work on the potential pharmacological activities in preventing diabetes complications as mentioned in the abstract.

 RESPOND:

·         The preliminary antioxidative and enzyme inhibitory activities were studied to determine the pharmacological potential of this plant especially in the aspect of its usefulness in the treatment of hyperglycemia in which digestive enzyme inhibition is one of the crucial ways of determining the anti-hyperglycemic activity of a plant.

5.      The interventional timing of each sample should be elaborated in the method of in vitro α-amylase inhibitory assay, in vitro α-glucosidase inhibitory assay, and ABTS assay.

 RESPOND:

·         Different concentrations (7.81-1000 µg/mL) were used in all the in vitro assays rather than timing.

6.      Why IC50 (μg/mL) was chosen as an approach to determine the inhibitory activity on digestive enzyme, but not the direct magnitude of reduction of enzyme activity?

 RESPOND

·         The protocols followed in determining the inhibition of α-amylase and α-glucosidase used IC50 to determine the inhibitory activity of all the samples.

Reviewer 3 Report

The presented here information are interesting but the way they are presented is unacceptable. While reading the chapter „Results and discussion” I had the impression that I was reading the methodology.

The authors should start the article from the beginning. First, a few sentences about the next stages of research such as: extraction, determination of the activity of individual extracts, selection of the best one. Then division into factions, testing them. Finally, the isolation of specific compounds.

The results should be presented in the same order.

Table.

First a short introduction, then a table, then a description of what follows from this table.

At the beginning of chapter 2.2 there is no information about fractions. What is it?

Extracts - why and for how long these solvents were used.

Can the TFC value be minus 468?

Table 4 – were is mentioned value R.

Fig. 3 - no description of TLC plates.

The final summary of results obtained from various measurements is missing. There are no conclusions resulting from the presented results.

Supplementary information

Presented here spectra are too small, they should be enlarged on the whole page.

Description of 1H NMR is not correct.

the multiplet must be given in the range, eg 2.01-2.06

what is singlet at 1.26?

signal coming from CH2-6 is not a doublet, but singlet

signals coming from H-1’ and H-6’ should be triplets

signals coming from 2’, 3’, 4’, 5’ can not be described as coming from 8 protons each

where are described multiplets 1.39-1.43 and 1.63-1.73?

Author Response

1.      The authors should start the article from the beginning. First, a few sentences about the next stages of research such as: extraction, determination of the activity of individual extracts, selection of the best one. Then division into factions, testing them. Finally, the isolation of specific compounds.

Respond

·         The methodology and results have been re-arranged and presented in the following sequence; plant preparation and extraction, fractionation of the active extract, determination of TPC and TFC, antioxidant activity, digestive enzyme inhibitory activity and finally isolation of the compounds. Now it is coherent.

·         Please see Pages 4-19. Changes highlighted in purple.

2.      First a short introduction, then a table, then a description of what follows from this table.

 Respond

·         Tables have been re arranged

·         Tables 1-5

3.      At the beginning of chapter 2.2 there is no information about fractions. What is it?

 Respond

·         Information about the fraction have been provided

·         Please see Page 3, line 44 and 45, line 207. Changes highlighted in purple

4.      Extracts - why and for how long these solvents were used.

 Respond

·         Chloroform: methanol solvents mixture was used because it gave the best yield based on the previous extraction done on the same plant’s stem bark. Gradient mixtures of the solvents were used from 9:1 to 6:1.

·         Please see Page 15, Line 308-309

5.      Can the TFC value be minus 468?

 Respond

·         Minus value of TFC indicates the absence of flavonoid in the sample.

6.      Table 4 – were is mentioned value R

 Respond

·         R-value was not included in the table, so R has been deleted in the footnote

·         Table 1, line 64

7.      Fig. 3 - no description of TLC plates

 Respond

·         TLC plates have been fully described

·         Figure 1, page 6, line99-100. Changes highlighted in green

8.      The final summary of results obtained from various measurements is missing. There are no conclusions resulting from the presented results.

 Respond

·         Summary has been included

·         Line 75, 117, 118, 129-130, 175-176

9.      Presented here spectra are too small, they should be enlarged on the whole page.

 Respond

·         Spectra have been enlarged to a whole page

·         Supplementary S1-S6

10.  Description of 1H NMR is not correct. the multiplet must be given in the range, eg 2.01-2.06

 Respond

·         1H NMR has been given in range

·         Supplementary S2 i and ii

11.  what is singlet at 1.26?

 Respond

·         Signal at 1.24-1.46 is a multiplet from H-4’ and 5’

·         Supplementary S2 (i)

12.  signal coming from CH2-6 is not a doublet, but singlet

 Respond

·         Signal from C-H at position 6 (of the phenolic ring) is a doublet J= 1.5 Hz

·         Table 3 supplementary S2 (ii)

13.  signals coming from H-1’ and H-6’ should be triplets

 Respond

·         Yes they are triplet

·         Supplementary S2 (i)

14.  signals coming from 2’, 3’, 4’, 5’ can not be described as coming from 8 protons each where are described multiplets 1.39-1.43 and 1.63-1.73?

 Respond

·         Signal from H-4’,5’ appeared as a multiplet at 1.39-1.43 ppm while 2’ and 3’ showed multiplet at 1.63-1.73 ppm

·         Supplementary S2 (i)

Reviewer 4 Report

In the study "Evaluation of Enzyme Inhibitory and Antioxidant Activities of Entada spiralis Stem Bark and Isolation of the Active Constituents" by Zakaria et al. the authors report the first study to date on the composition of this medicinal plant, particularly of its stem bark. While the methodology applied is adequate and the data obtained is properly presented, the authors have not had any luck in terms of novelty regarding the isolated products: 7 out of the 8 pure compounds reported here are well known phenolics, and the only one that wasn't, although correctly characterized, does not entail any particular structural or biological significance, all of which diminishes the urgency and novelty of this communication.

Regardless of the potential outcome of this submission, the authors should consider revising the following points:

- Biological activities of crude product mixtures can be given in ug/mL, since at this point of the analysis the composition of the mixture is unknown. However, for pure isolated products, whose molecular weight is known, IC50 values would be much more informative if they were expressed as mM/uM/nM units instead. 

- references should incorporate a recent example of outstanding inhibition of digestive enzymes by the flavonol glycoside Montbretin A (doi: 10.1038/nchembio.1865) 

- Fig. S5, the molecular structure should be redrawn, having into consideration the E configuration of the double bond in FEQ-2.

-  No supplementary information is provided to support IC50 values. Addition of curves or tables with actual data to the supp. info. is encouraged.

-  the use of the word "least" throughout the entire manuscript when the authors actually mean "lowest" is discouraged.

Author Response

1.      In the study "Evaluation of Enzyme Inhibitory and Antioxidant Activities of Entadaspiralis Stem Bark and Isolation of the Active Constituents" by Zakaria et al. the authors report the first study to date on the composition of this medicinal plant, particularly of its stem bark. While the methodology applied is adequate and the data obtained is properly presented, the authors have not had any luck in terms of novelty regarding the isolated products: 7 out of the 8 pure compounds reported here are well known phenolics, and the only one that wasn't, although correctly characterized, does not entail any particular structural or biological significance, all of which diminishes the urgency and novelty of this communication.

 RESPOND

·         Although 7 out of 8 compounds isolated have been previously reported from different plants but this is the first time of reporting the isolation of these compounds from E. spiralis. Despite the fact that FEQ-2 did not show any detectable amylase inhibition, it exhibited good radical scavenging and α-glucosidase inhibition.

2.      Biological activities of crude product mixtures can be given in ug/mL, since at this point of the analysis the composition of the mixture is unknown. However, for pure isolated products, whose molecular weight is known, IC50 values would be much more informative if they were expressed as mM/uM/nM units instead. 

 RESPOND

·         Results obtained from all the biological analysis were reported in one table to prevent unnecessary bulkiness of the article and therefore ug/mL was used to expressed all the IC50 of  all the samples.

3.      references should incorporate a recent example of outstanding inhibition of digestive enzymes by the flavonol glycoside Montbretin A (doi:10.1038/nchembio.1865)

 RESPOND

·         Reference has been incorporated.

·         Please see Page 3, lines 24 and 681, reference number 46, highlighted in green colour.

4.      Fig. S5, the molecular structure should be redrawn, having into consideration the E configuration of the double bond in FEQ-2.

 RESPOND

·         Figure S5 has been re-drawn

·         Please see Supplementary S5

5.      No supplementary information is provided to support IC50 values. Addition of curves or tables with actual data to the supp. info. is encouraged.

 RESPOND

·         The raw data for all the biological activities can be made available upon request.

6.      the use of the word "least" throughout the entire manuscript when the authors actually mean "lowest" is discouraged.

 RESPOND

·         “least has been changed to lowest

·         Please see Pages 5 and 7, lines 71, 109, 110, 116, 120, 146, 147, 183 and 193. Changes highlighted in green.

Round  2

Reviewer 2 Report

1)  2) Extensive format check is needed throughout the manuscript.

Author Response

Comments and Suggestions for Authors

The manuscript has been revised.  Some further minor revisions are needed:

1)  In Table 2: Regarding the sample F3, please double check the value “215 ± 86 ± 16.62”.

RESPOND:

The value of F3 has been corrected in Table 2

2) Extensive format check is needed throughout the manuscript.

RESPOND:

The manuscript has been carefully checked and thoroughly formatted to improve it further.

Reviewer 3 Report

The manuscript has been improved. The authors answered all of my questions. I have only minor comments regarding the description of NMR spectra of the compound FEQ-2.

In Table 3:

signal coming from 2’ proton should be described as 1.63-1.73, 2H, m

signal coming from 3’ proton should be described as 1.63-1.73, 2H, m

signal coming from 4’ proton should be described as 1.39-1.43, 2H, m

signal coming from 5’ proton should be described as 1.39-1.43, 2H, m

Figure S2

S2 (i) - the range for signals coming from the 2'-5 'protons indicated in the spectrum should be in accordance with the range given in Table 3

S2(ii):

instead of H-6 should be H-2

instead of H-2 should be H-6

instead of H-5 should be H-3

Author Response

Comments and Suggestions for Authors

The manuscript has been improved. The authors answered all of my questions. I have only minor comments regarding the description of NMR spectra of the compound FEQ-2.

In Table 4:

signal coming from 2’ proton should be described as 1.63-1.73, 2H, m

signal coming from 3’ proton should be described as 1.63-1.73, 2H, m

signal coming from 4’ proton should be described as 1.39-1.43, 2H, m

signal coming from 5’ proton should be described as 1.39-1.43, 2H, m

RESPOND:

The following modifications have been made in Table 4

signal coming from 2' proton has been described as 1.63-1.73, 2H, m

signal coming from 3' proton has been described as 1.63-1.73, 2H, m

signal coming from 4' proton has been described 1.35-1.43, 2H, m

signal coming from 5' proton has been described as 1.35-1.43, 2H, m

Figure S2

S2 (i) - the range for signals coming from the 2'-5 'protons indicated in the spectrum should be in accordance with the range given in Table 3

S2(ii):

instead of H-6 should be H-2

instead of H-2 should be H-6

instead of H-5 should be H-3

RESPOND:

For supplementary file, in Figure S 2 (i) the range for signals coming from the 2'-5'protons indicated in the spectrum have been labeled in accordance with the range given in Table 4.In Figure S2(ii),

H-6 has been re-labeled to be H-2

H-2 has been re-labeled to be H-6

H-5 has been re-labeled to be H-3

Reviewer 4 Report

In this new version of their manuscript, the authors have incorporated several changes that result in a more appealing study, despite the fact that most of the modifications are of cosmetic nature. However, this reviewer must insist on the appropriateness of splitting the biological activity table in two, one for crude materials/fractions, in which IC50 values can be expressed in ug/mL, and a second one for purified compounds, in which IC50 values are to be expressed in molar concentrations. 

Also, the addition of reference 46 should be accompanied at least by a mention of the mode of inhibition of these flavonoids.

Author Response

Comments and Suggestions for Authors

In this new version of their manuscript, the authors have incorporated several changes that result in a more appealing study, despite the fact that most of the modifications are of cosmetic nature. However, this reviewer must insist on the appropriateness of splitting the biological activity table in two, one for crude materials/fractions, in which IC50 values can be expressed in ug/mL, and a second one for purified compounds, in which IC50 values are to be expressed in molar concentrations. 

RESPOND:

Table 2 has been splitted; Table 2 includes IC50 of the biological activities of the crude extracts and methanol fractions in µg/mL while Table 3 contains the IC50 of biological activities of the isolated compounds expressed in µmol/ L.

Also, the addition of reference 46 should be accompanied at least by a mention of the mode of inhibition of these flavonoids.

RESPOND:

The literature of mode of the inhibition of flavonol glycoside (montbretin A) on HPA has been elaborated in the introduction (Page 3, L 24-26).